# Clinical Application of Virtual Reality for Upper Limb Motor Rehabilitation in Stroke: Review of Technologies and Clinical Evidence

**DOI:** 10.3390/jcm9103369

**Published:** 2020-10-21

**Authors:** Won-Seok Kim, Sungmin Cho, Jeonghun Ku, Yuhee Kim, Kiwon Lee, Han-Jeong Hwang, Nam-Jong Paik

**Affiliations:** 1Department of Rehabilitation Medicine, Seoul National University College of Medicine, Seoul National University Bundang Hospital, Seongnam-si, Gyeonggi-do 13620, Korea; njpaik@snu.ac.kr; 2Delvine Inc., Seoul 08788, Korea; cho@delvine.co.kr; 3Department of Biomedical Engineering, College of Medicine, Keimyung University, Daegu 42601, Korea; 4Ybrain Research Institute, Seongnam-si, Gyeonggi-do 13449, Korea; yuhee.kim@ybrain.com (Y.K.); kiwon.lee@ybrain.com (K.L.); 5Department of Electronics and Information, Korea University, Sejong 30019, Korea; hwanghj@korea.ac.kr

**Keywords:** virtual reality, stroke, rehabilitation, hemiplegia, recovery of function, neuronal plasticity, sensor

## Abstract

Neurorehabilitation for stroke is important for upper limb motor recovery. Conventional rehabilitation such as occupational therapy has been used, but novel technologies are expected to open new opportunities for better recovery. Virtual reality (VR) is a technology with a set of informatics that provides interactive environments to patients. VR can enhance neuroplasticity and recovery after a stroke by providing more intensive, repetitive, and engaging training due to several advantages, including: (1) tasks with various difficulty levels for rehabilitation, (2) augmented real-time feedback, (3) more immersive and engaging experiences, (4) more standardized rehabilitation, and (5) safe simulation of real-world activities of daily living. In this comprehensive narrative review of the application of VR in motor rehabilitation after stroke, mainly for the upper limbs, we cover: (1) the technologies used in VR rehabilitation, including sensors; (2) the clinical application of and evidence for VR in stroke rehabilitation; and (3) considerations for VR application in stroke rehabilitation. Meta-analyses for upper limb VR rehabilitation after stroke were identified by an online search of Ovid-MEDLINE, Ovid-EMBASE, the Cochrane Library, and KoreaMed. We expect that this review will provide insights into successful clinical applications or trials of VR for motor rehabilitation after stroke.

## 1. Introduction

Stroke is one of the leading causes of disability and socioeconomic burden worldwide [1]. Although the age-standardized stroke incidence has decreased in most regions, the growth of aging populations, who are at risk of stroke, may lead to an increase in the crude incidence of stroke [2]. According to a policy statement by an American Heart Association working group, approximately 4% of US adults will have a stroke by 2030 [3]. Stroke-related mortality has shown a remarkable decline due to better management in the acute phase, which means there are more people living with disabilities after stroke [1,3].

Upper limb hemiparesis is one of the most common impairments after stroke [4] and is associated with activity limitation and a worse quality of life [5,6,7]. Therefore, adequate recovery of upper limb weakness is necessary. Spontaneous motor recovery occurring up to one year after stroke can be accelerated with active rehabilitation strategies [8,9]. However, the effects of conventional rehabilitation modalities are limited and novel therapeutic approaches are required [10].

Virtual rehabilitation using virtual reality (VR) technology is a novel promising modality for motor rehabilitation after stroke [11] that can add beneficial components to current rehabilitation strategies. Considering motor learning theory, task-oriented, intensive (that is, more doses and movements), and repetitive training is essential for promoting neuroplasticity and thereby, motor recovery (Figure 1) [12]. Several advantages of virtual rehabilitation can be suggested in terms of rehabilitation intensity and motivation. VR can motivate patients’ participation by increasing enjoyment and gamification—“the process of adding game-design elements and game principles to something (e.g., task) so as to encourage participation”—thereby increasing task repetition (intensity) [13,14,15]. Flexible and individualized rehabilitation design is possible according to the patient’s motor impairment, which makes the step-by-step approach possible. A low-cost virtual rehabilitation system can be used as an adjunctive therapy to conventional rehabilitation, with less direct supervision by a therapist [16], and it can also be considered for use as a tele- or home-based rehabilitation tool [17]. Functional assessment and digital tracking of patients’ progress is possible using motion sensors combined with VR systems for rehabilitation [18].

In this comprehensive narrative review of the application of VR in motor rehabilitation after stroke, we will cover (1) the technologies used in VR rehabilitation including sensors, haptic devices, and VR displays; (2) the clinical application and evidence for VR in motor rehabilitation in stroke; and (3) considerations for VR application in stroke rehabilitation. We expect that this review will provide insights into successful clinical applications or trials of VR for motor rehabilitation after stroke.

## 2. Technologies Used in VR Rehabilitation

### 2.1. Definition of VR

VR technology can give users the experience of being surrounded by a computer-generated world. With VR, users experience inclusive and extensive surroundings, with vivid illusions of a virtual computer generated environment in which both realistic and unrealistic events can occur. So, users can interact as though they are in a real environment and may not even recognize that they are existing in a virtual environment [19]. Therefore, in VR, participants can be fully immersed in the surrounding virtual environment and interact naturally with virtual objects in the virtual world [20]. Because VR content responds to a user’s movements in a natural and valid manner, such as showing the corresponding scene on the display when the user looks at it, the interaction evokes a feeling of existing in a virtual environment, which is referred to as “presence.” Moreover, control of the avatar’s body movements by those of the user can even induce a feeling of ownership in which the user regards the avatar’s body parts as surrogates of their own, a phenomenon called “virtual embodiment” [21]. Based on these factors, users are fully immersed, which allows them to experience where they are and what they do there in a way that is similar to real/lived experience.

### 2.2. Non-Immersive and Immersive VR

Non-immersive VR allows users to experience a virtual environment as observers and interact with the virtual environment by using devices that cannot fully overwhelm sensory perceptions [22], which results in a lesser feeling of immersion in the virtual world. Non-immersive VR systems are mainly characterized by users’ ability to control their surroundings while perceiving stimuli around them, such as sounds, visuals, and haptics. Non-immersive VR systems are primarily based on a computer or video game console, flat screen, or monitor and input devices such as keyboards, mice, and controllers. Non-immersive VR systems can also use other physical input devices, such as racing wheels, pedals, and speed shifters, to augment users’ realistic experiences. Using various input devices, users can interact with VR content on a display. To enhance the level of immersion, some non-immersive VR systems provide a first-person view for users to associate themselves with their virtual avatar. To allow users to perceive objects as being 3D, stereoscopic vision technology, with which stereo images are provided so that each eye of the user, who wears special goggles, receives the same scene but from a slightly different angle, could be used, which would allow the user to feel the third dimension from a 2D monitor or screen [23].

Immersive VR, on the other hand, improves the feeling of presence, enabling people to feel more like they are actually in the virtual environment, which means that users are more likely to interact with the stimuli given by the computer and related devices providing visual, auditory, and haptic sensations. The main goal of immersive VR is to make it possible for users to experience the illusion of being in the computer-generated environment rather than the real-world environment. By wearing a head-mounted display (HMD), tracking devices, haptic devices, and data gloves and by using wireless controllers, users can be placed in virtual environments and interact with a computer-generated world. However, the real world has a greater variety of senses including smell, taste, the feeling of warm and cold, etc., which may increase the gap between the virtual and real worlds. These could be further covered by complete immersive VR, but the need for a sophisticated artificial stimulator to provide variable sensations may require more space and have a higher cost. HMD-based immersive VR could also be enriched using physical objects or devices placed in the physical space by tracking their positions precisely in relation to where the user stands. By using this paradigm, the user could perceive the texture or temperature of objects without any awkwardness when touching it because the physical object is tracked to be placed at the same position as that in the virtual space; thus, the user touches the physical object when they touch the virtual object. Another issue to overcome is that the user must be placed in a limited space; therefore, their walking area is constrained. Using a VR treadmill allows users to physically walk or run toward any place in a virtual environment by solving two problems: realistic synchronized simulation of the user’s walk and no requirement for a large space. The Cave Automatic Virtual Environment (CAVE) has been introduced as another way to provide visual information for immersive VR, instead of using an HMD [24]. CAVE uses six large walls on which scenes are displayed so that the participant can be placed in the CAVE and experience the surrounding virtual environment with a large field of view.

Immersive and non-immersive environments can be better differentiated by their level of immersion. Immersive VR strengthens the level of immersion because less mental effort is required to be immersed in the virtual environment since the hardware systems cover most sensory perceptions. In contrast, non-immersive VR requires more mental effort to be immersed in the virtual environment. Therefore, non-immersive VR may reduce the level of spatial presence, which is defined as “the sense of being in an environment” [25,26].

### 2.3. Technologies for Motion Tracking and Feedback for Virtual Rehabilitation

Virtual rehabilitation is a method of rehabilitation via gamification through synchronization with software content or by providing a motion guide. Various studies have been conducted to investigate the application of VR for upper limb rehabilitation (Table 1).

For motor rehabilitation of limbs, the patient’s body part must be captured by motion tracking sensors and synchronously transferred to an object in VR. Sensors to track the patient’s motion are mandatory for movement visualization and can be selected from a mouse and joystick, depth-sensing cameras, electromagnetic sensors, inertial sensors, bending sensors, data gloves, and so on. The sensor performance is important to precisely track the motion, but the subjective perception and preferences are also important factors to be considered, in addition to cost [27].

A sensor technology that recognizes motion is essential for virtual rehabilitation. Such technologies are divided into wearable and nonwearable devices that recognize upper limb rehabilitation motions. Nonwearable devices are further divided into those using a vision sensor and those using a robot-based controller or a controller with three degrees of freedom (DOF) either alone or in combination. Wearable devices are usually divided into those using data gloves and those using an exoskeleton. Some studies have used both types together. Sensing using cameras in nonwearable devices has recently changed from tracking markers or color patches using webcams to tracking body or hand signals through depth sensing methods. In this way, the users’ movements are sensed within a limited space without obstacles. With wearable devices, the sensor is attached to collect high frequency data and force or torque can be tracked as well as position and movement.

Most studies have primarily used visual and auditory feedback through content and some studies have applied tactile and force feedback (which are haptic feedback). We divided the studies into visuomotor and visuohaptic feedback. Visuomotor feedback provides visual information by applying measured movements through sensors to content in real time. Visuohaptic feedback refers to providing haptic feedback with visual information. Haptic feedback could be divided into tactile or force feedback, depending on whether resistance is present. Tactile feedback provides feedback to users through the sense of touch using vibration, skin deformation, or small forces. Force feedback, or kinesthetic force feedback, simulates real-world physical touch using motorized motion or resistance rather than by fine touch [28]. Research on virtual rehabilitation can be categorized according to the use of fine motor tracking during upper limb rehabilitation, which is distinguished by the use of wearable or nonwearable devices. In the case of camera methods among nonwearable devices, hand tracking is possible with a HMD for VR, and this has been released as a commercial product (e.g., Oculus Quest, Facebook Technologies, LLC, Menlo Park, CA, USA). The sensor types used in previous studies are summarized in Figure 2.

### 2.4. Examples of Commercialized VR Upper Limb Rehabilitation Systems

Commercialized virtual rehabilitation devices that can provide gamification through content or rehabilitation guides are similar to those used in VR rehabilitation research, but they are simplified and more focused on ease of use. Sensing methods are divided into wearable and nonwearable methods using cameras, joysticks, and robots (Table 2).

## 3. Clinical Evidence and Considerations for VR in Motor Rehabilitation after Stroke

### 3.1. Literature Search

Studies for upper limb VR rehabilitation after stroke were identified by an online search of Ovid-MEDLINE, Ovid-EMBASE, the Cochrane Library, and KoreaMed on 18 June 2020. The search queries are presented in Appendix A. Titles and abstracts were reviewed for screening by Y.K. and non-English papers, animal studies, commentaries, case series, narratives, book chapters, editorials, nonsystematic reviews, and conference papers were excluded. Duplicated publications between databases were also excluded. A total of 339 studies were included for the full text review and Y.K. and W.S.K. selected systematic reviews and meta-analyses for review. Six meta-analyses were included for our evidence summary [63,64,65,66,67,68].

### 3.2. Clinical Evidence

The general characteristics of the included meta-analyses are presented in Table 3. Two studies included randomized controlled trials (RCTs) and quasi-randomized controlled trials [64,68] and three other studies only included RCTs [63,65,67]. Karamians et al. included RCTs and prospective studies [66]. The number of studies and participants included in each meta-analysis ranged from 21 to 72 and 562 to 2470, respectively.

The interventions were VR rehabilitation and the controls were either conventional therapy (dose-matched or not) or no intervention (Table 4). VR rehabilitation included rehabilitations using both custom-built virtual environments and commercial video gaming consoles (e.g., Nintendo Wii or Xbox Kinect) in the selected meta-analyses. Outcomes were usually the composite outcomes of upper limb function or activities and Karamians et al. only included studies using one of the following outcome measures: Fugl-Meyer Assessment (FMA), Wolf Motor Function Test (WMFT), and Action Research Arm Test (ARAT) [66]. Mekbib et al. only included studies using one of the three following outcomes: FMA, Box and Block Test (BBT), and Motor Activity Log (MAL) [65]. Methodological quality of the included meta-analyses was assessed using the Assessment of Multiple Systematic Reviews (AMSTAR 2) instrument by two authors (S.C. and W.S.K.) and was categorized as high, moderate, low, or critically low [69]. Any disagreements were resolved through the discussion for consensus. Most of the included meta-analyses showed moderate to high methodological quality (Table 4).

In one high-quality meta-analysis from 2014, VR rehabilitation showed better improvements in body function (standardized mean difference (SMD) = 0.43, 95% confidence interval (CI) = 0.22 to 0.64) and activities (SMD = 0.54, 95% CI = 0.28 to 0.81) when compared to conventional therapy [64]. However, the commercially available gaming failed to show a significant beneficial effect due to the small number of studies (Table 4). In a recent Cochrane systematic review with high methodological quality, VR rehabilitation for the composite outcome of upper limb function (primary outcome) was not superior to conventional therapy, but upper limb function measured by FMA was significantly improved in VR rehabilitation (SMD = 2.85, 95% CI = 1.06 to 4.65) [68]. When VR rehabilitation was applied in addition to conventional therapy, VR rehabilitation showed significant beneficial effects on the composite outcome of upper limb function (SMD = 0.49, 95 CI = 0.21 to 0.77). Two metanalyses by Aminov et al. [63] and Lee et al. [67] also showed similar moderate effect sizes for upper limb function in VR rehabilitation (Table 4). Mekbib et al. [65] only included RCTs using dose-matched conventional therapy and calculated the mean differences of FMA, BBT, and MAL, which all represent upper limb function. Although VR showed better improvements in all outcomes when compared to conventional therapy, they were less than the minimal clinically important difference [70,71].

## 4. Considerations for VR Application in Stroke Rehabilitation

### 4.1. HMDs and Motion Sickness

HMDs give users a more immersive experience in a 3D artificial world and allow interaction with virtual objects using motion tracking sensors. Considering the therapy time and active motion during the rehabilitation, the HMD must be light, comfortable to wear, positioned stably on the head, and cool enough during operation (HMD typically generate heat). The HMD may also benefit from being wireless (with enough battery life). Although VR rehabilitation can induce eye strain or physical fatigue during extended therapy, the most common issue to overcome is motion sickness. Motion sickness can be elicited when there is a lag in processing the visual response to user input interactions, resulting in conflicting signals to the brain from the eyes, vestibular systems in the inner ear, and proprioceptive sensory receptors (sensory conflict theory) [72]. Motion sickness can be affected by the system (e.g., head tracking, rendering, field of view, optics); application and user interaction (lack of controlling visual motion, visual acceleration or deceleration, longer duration of VR experience, frequent head movement during VR play); and individual perceptual factors (age, motion sickness history, lack of VR experience). The following approaches can be employed to reduce motion sickness when designing VR rehabilitation programs [73]: “(1) to make patients actively control their view points and be responsible for initiating movement, (2) to avoid or limit linear or angular accelerations or decelerations without corresponding vestibular stimulation, (3) to display visual indicators or motion trajectories, (4) to display visual cues that remain stable as the patient moves, and (5) to perform dynamic blurring of unimportant areas.”

### 4.2. Differences in Movements in VR

The movement kinematics of the upper limb in patients with stroke differ between VR and real environments. Viau et al. reported that patients with hemiparesis used less wrist extension and more elbow extension at the end of the placing phase during reaching, grasping, and performing tasks in VR than in a real environment [74]. Similarly, several studies using reaching tasks also demonstrated that the movements in VR using HMDs were slower than those in the real environment and that spatial and temporal kinematics differ between VR and real environments [75,76,77]. Lott et al. reported that the range of the center of pressure during reaching in standing (usually used for balance training) was different between real environments, non-immersive VR with 2D flat-screen displays, and immersive VR with HMDs [78]. Considering the rehabilitation purpose of improving independence in real-world living, these different movement kinematics can affect the transfer of learning in VR to real environments and therefore must be considered when designing a VR-based rehabilitation program.

### 4.3. Transfer of Learning in VR to the Real World

The transfer of improved function after rehabilitation to the performance of activities of daily living is important in upper limb rehabilitation after stroke. Constraint-induced movement therapy (CIMT) comprises repetitive tasks/shaping practice with constraint of the hemiparetic upper limb, emphasizing the transfer package to foster compliance and use of the hemiparetic upper limb in the real world as a key component to improve function following CIMT [79]. Therefore, the transfer of novel rehabilitation therapeutic approaches based on repetitive movements to the real environment, such as robot-assisted arm rehabilitation [80] and VR-based rehabilitation [81], is an important issue to be discussed.

The transfer of learning effects in VR to real environments is inconclusive. Rose et al. showed that the effect of simple sensorimotor task training is comparable between VR and real environments [82]. However, several recent studies have shown that training in VR did not translate to better performance in the real environment [83,84,85,86]. In the virtual BBT simulated using a 2D flat screen and depth-sensing camera, the number of boxes moved in VR showed good correlation (a high correlation coefficient) with that in the real BBT, but the actual number of boxes moved was much less in the VR condition [33]. The weak transfer of effects from VR to real environments may be associated with different sensory-motor symptoms and spatiotemporal organization, especially the differences in depth perception in VR during upper limb rehabilitation (reach, touch, grasp, and release tasks). Although an HMD improves depth perception compared to a 2D flat screen display [87], further improvements in VR depth perception, and thereby fidelity, is needed. Possible strategies include object occlusion; effects of lighting and shadow; color shading; and relative scaling of objects by considering depth, perspective projection, and motion parallax [88]. Other methods to improve the interaction can be visual (e.g., color change) or auditory feedback when touching objects in VR. Haptic feedback can further improve the interaction and thereby the fidelity of the VR training. Ebrahimi et al. demonstrated that the errors and time to complete the task during reaching and pointing tasks using a stylus in immersive VR with a HMD were decreased with the addition of visuohaptic feedback compared to the condition without it [89]. It has also been suggested that matching the VR interaction dimensions with the control dimension of the task in the real world could improve the transfer of the VR rehabilitation effect [90].

### 4.4. Gamification

Gamification has been broadly and clearly defined as the “use of game design principles in non-game contexts” by Deterding et al. [91]. Gamification of VR-based rehabilitation systems can motivate patients to participate in rehabilitation actively with enjoyment, which could lead to more movements of the hemiparetic arm and better recovery [92]. The strategies to apply gamification to virtual rehabilitation design have been thoroughly reviewed by Charles et al. [88] and Mubin et al. [93]. Briefly, the VR rehabilitation system must give the patients clear feedback for meaningful play, such as the therapist’s verbal and emotional encouragement, with a clear goal to be achieved during the occupational therapy. The difficulty level or challenge during the rehabilitation game should be adapted according to the patient’s ability to facilitate meaningful play and handle failures [94,95] as patients with stroke may experience multiple failures and can be frustrated during upper limb rehabilitation due to motor impairments. Various types of feedback, including visual, auditory, and haptic feedback, can be applied and approaches to possibly promote motor learning should be considered (inducing variability of tasks, amplification of visual errors, and manipulating task physics for implicit behavioral guidance) [81].

### 4.5. Barriers

In addition to the barriers caused by patients (physical and cognitive disabilities, low adoptability, and compliance to technology), it has been suggested that there are also barriers at the therapist level, which can lead to underuse of VR rehabilitation [96]. Glegg et al. recently reviewed the barriers and facilitators influencing the adoption of VR rehabilitation, which include “the ability to grade the degree of training, transfer of training to real life, knowledge about how to operate the VR clinically, therapist self-efficacy and perceived ease of use, technical and treatment space issues, access to the technology, and time to learn practice for VR rehabilitation” [97]. They gave three recommendations to promote the use of VR rehabilitation, which were “(1) enhance collaboration, (2) ensure knowledge transfer interventions are system- and context-specific, and (3) optimize VR effectiveness through an evidence-based approach” [97].

## 5. Combinational Approaches with VR in Stroke Rehabilitation

Neuroplasticity is the ability of the human brain to adapt to certain experiences, environments, and extreme changes, including brain damage [98,99,100]. Several novel therapeutic approaches to enhance neuroplasticity can be considered as combinational approaches to VR rehabilitation.

The brain-computer interface (BCI) is one method used to improve neuroplasticity after stroke; it is based on motor imagery, which is defined as the mental simulation of a kinesthetic movement. The BCI provides sensory feedback of ongoing sensorimotor brain activities, thereby enabling stroke survivors to self-modulate their sensorimotor brain activities [101]. BCI for motor rehabilitation involves the recording and decoding of brain signals generated in the sensorimotor cortex areas. The recorded brain signals can be used (1) to objectify and strengthen motor imagery-based training by providing stroke patients with real-time feedback on an imagined motor task; (2) to generate a desired motor task by producing a command to control external rehabilitative tools, such as functional electrical stimulation, robotic orthoses attached to the patient’s limb, or VR; and (3) to understand cerebral reorganizations of lesioned areas by quantifying plasticity-induced changes in brain networks and power spectra in motor-related frequency bands (i.e., alpha and beta) [102]. A previous meta-analysis reported that BCI had an SMD of 0.79, which represented a medium to large effect size comparable with those of conventional rehabilitation therapy such as CIMT (SMD = 0.33), mirror therapy (SMD = 0.61), and mental practice (SMD = 0.62) [101]. Pichiorri et al. showed that BCI combined with VR may further improve upper limb rehabilitation outcomes and may be used to predict motor outcomes by analyzing brain activity in patients with stroke [103]. A more recent study also demonstrated the clinical feasibility of using a combination of BCI and VR in post-stroke motor rehabilitation and confirmed that this combinatory method may benefit patients with severe motor impairments who have little ability for volitional movement [104].

Another novel strategy to increase neuroplasticity using noninvasive brain stimulation, such as transcranial magnetic stimulation (TMS) and transcranial direct current stimulation (tDCS), has also been suggested by various researchers [105,106,107]. Noninvasive brain stimulation methods can be used to (1) enhance the ipsilesional brain activity by high-frequency rTMS [108] or anodal tDCS [109,110]; (2) inhibit contralesional brain activity by low-frequency rTMS [111,112] or cathodal tDCS [113]; (3) produce an additive effect by simultaneously applying anodal tDCS over the ipsilesional area and cathodal tDCS over the contralesional area, referred to as bihemispheric tDCS [114]; or (4) modulate the somatosensory input from nerve fibers to the brain [115,116]. These noninvasive brain stimulation methods have been reported to have acceptable tolerability and safety with no significant adverse effects in various populations, including patients with stroke [117,118]. Several studies have shown a positive effect of combinational approaches (noninvasive brain stimulation plus VR-based rehabilitation) in patients following stroke [119,120,121].

Together with BCI and noninvasive brain stimulation, a telerehabilitation approach may also be important for motor rehabilitation after stroke in terms of better accessibility and prolonged usage at home. Telerehabilitation, by definition, can provide the inputs of multidisciplinary skilled personnel for rehabilitation, including physiatrists, physiotherapists, and occupational therapists, which are often unavailable at home or challenged by transportation restrictions for disabled patients [122]. A recent Cochrane systematic review showed moderate-quality evidence that there was no difference in activities of daily living in patients with stroke between those who received telerehabilitation and those who received usual care (SMD = −0.00, 95% CI = −0.15 to 0.15) [123]. There was also low-quality evidence of no difference in upper limb functions between the use of a computer program to remotely retrain upper limb function and in-person therapy (mean difference = 1.23, 95% CI = −2.17 to 4.64) [123]. Several studies have shown that VR based telerehabilitation can be used for motor rehabilitation of upper extremity functions with improvements in FMA of the upper extremity, Brunnstrom stage, manual muscle test, and action research arm test [124,125].

## 6. Summary

VR-based rehabilitation is a promising tool to actively engage patients in the rehabilitation program and can lead to better motor recovery. Although current clinical evidence shows that VR-based rehabilitation is beneficial as an adjunct therapy to convenient rehabilitation therapy, the interventions in the studies included in the meta analyses were heterogeneous and it is unclear who benefits more from VR rehabilitation (e.g., severity of impairment, time since onset of stroke) and what type of VR (e.g., immersive vs. non-immersive) and feedback is more effective. Further research including large well-designed RCTs to find the factors influencing the effects of VR rehabilitation are required.

To improve the efficacy of VR-based rehabilitation, VR rehabilitation is designed to improve the transfer of VR training to real environments, gamification, and feedback to promote active patient participation and neuroplasticity is necessary. The user interface and user experience must be designed to be more user-friendly to patients and therapists, considering both the patient’s physical and cognitive impairments and therapists’ needs. VR can be integrated into novel therapeutic modalities that can enhance neuroplasticity (e.g., BCI and noninvasive brain stimulation) and is expected to induce better recovery by combinational approaches, which warrant further investigation.

## Figures and Tables

**Figure 1 jcm-09-03369-f001:**
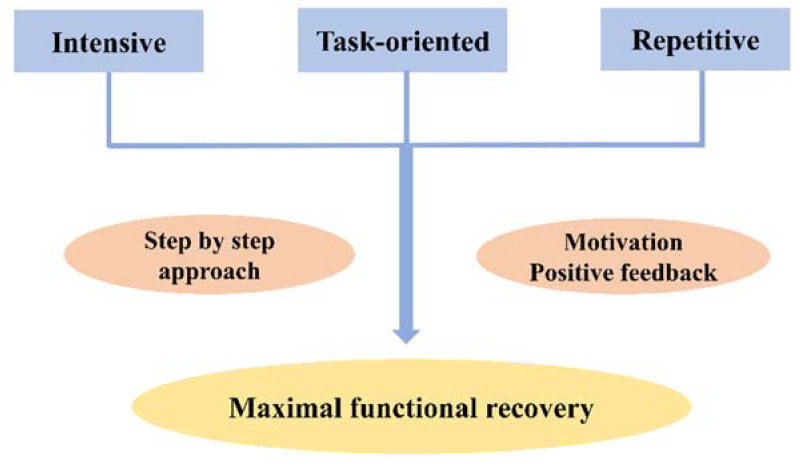
Approaches to promote neural plasticity.

**Figure 2 jcm-09-03369-f002:**
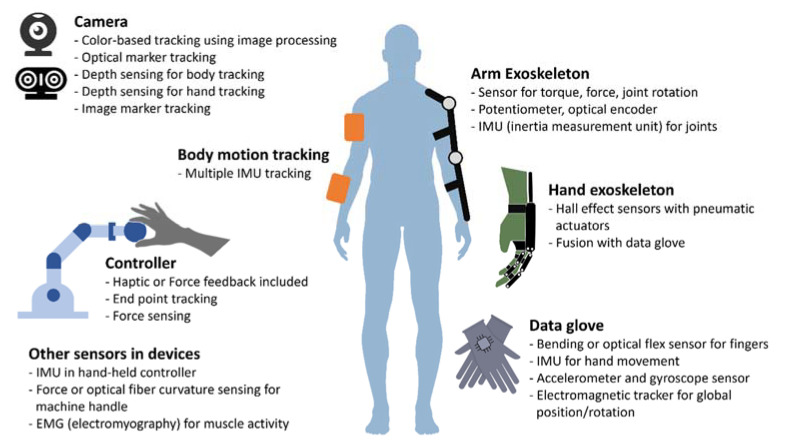
Classification of sensor types used in virtual rehabilitation for upper limb rehabilitation.

**Table 1 jcm-09-03369-t001:** Virtual rehabilitation studies for upper limb rehabilitation.

Study	Sensor Type	Sensor Type (Detail)	Feedback Type	VR Type	Rehabilitation Part
**Visuomotor Feedback**					
Dimbwadyo-Terrer et al., 2016 [29]	Wearable	Data glove	V	NI	Arm, hand, finger
Crosbie et al., 2012 [30]	Wearable	Data glove, motion tracking sensor	V, A	I	Arm, hand, finger
Calabrò et al., 2019 [31]	Wearable	End-effector hand exoskeleton	V, A	NI	Finger
Kiper et al., 2018 [32]	Wearable	Electromagnetic sensor	V	NI	Arm, hand
Cho et al., 2016 [33]	Nonwearable	Motion-sensing camera (depth sensing, hand tracking)	V, A	NI	Arm, hand, finger
Askin et al., 2018 [34]	Nonwearable	Motion-sensing Camera (body tracking with depth sensing)	V	NI	Arm
Faria et al., 2018 [35]	Nonwearable	Marker-based tracking with webcam	V	NI	Arm, hand
Lee et al., 2018 [36]	Nonwearable	Controller for paddling movement (canoe-like apparatus)	V, A	NI	Arm, hand
Sucar et al., 2014 [37]	Nonwearable	Pressure sensor in custom gripper, colored object tracking with webcam	V, A	NI	Arm, hand
Ballester et al., 2017 [38]	Wearable, nonwearable	Motion-sensing camera (depth sensing, body tracking), data glove	V	NI	Arm, hand, finger
Sampson et al., 2012 [39]	Wearable, nonwearable	Colored object tracking with webcam	V	NI	Arm
Xin et al., 2014 [40]	Wearable, nonwearable	Motion-sensing camera (body tracking with depth sensing), EMG sensing	V	I	Arm
**Visuohaptic Feedback**					
Feintuch et al., 2006 [41]	Wearable	Colored glove tracking with webcam	T, V, A	NI	Arm
Popescu et al., 2000 [42]	Wearable	Non-contact position sensors	F, V, A	NI	Hand, finger
Prisco et al., 1998 [43]	Wearable	Glove with electromagnetic measurements, torque/force and joint rotation sensing in arm exoskeleton	F, V, A	I	Arm, hand, finger
Alamri et al., 2008 [44], Kayyali et al., 2007 [45]	Wearable	Data glove with hand exoskeleton	F, V	NI	Arm, hand, finger
Adamovich et al., 2009 [46]	Wearable	Data glove with hand exoskeleton	F, V, A	NI	Arm, hand, finger
Molier et al., 2011 [47]	Wearable	Potentiometer and optical encoder in arm exoskeleton	F, V, A	NI	Arm, hand
Jack et al., 2001 [48] Merians et al., 2002 [13]	Wearable	Non-contact position sensor, data glove	F, V, A	NI	Hand, finger
Wille et al., 2009 [49]	Wearable	Data glove, accelerometers, and magnetometers	T, V	NI	Arm, hand, finger
Connelly et al., 2009 [50]	Wearable	Data glove, magnetic tracker for head tracking	T, V, A	I	Hand, finger
Huang et al., 2017 [51]	Wearable	Position and force sensor in hand rehabilitation robot	V, A	I	Finger
Pignolo et al., 2012 [52]	Wearable	Optical encoder in arm exoskeleton	V, A	I	Arm
Andaluz et al., 2016 [53], Bardorfer et al., 2001 [54]	Nonwearable	3D controller including buttons	F, V	NI	Arm, hand, finger
Broeren et al., 2004 [55]	Nonwearable	3D controller	F, V	SI	Hand, finger
Adamovich et al., 2009 [56]	Nonwearable	Force sensor in 3 DOF admittance-controlled robot	F, V	NI	Arm, hand
Merians et al., 2011 [57]	Nonwearable	Data glove, optical fiber curvature sensor, force sensor in 3 DOF admittance-controlled robot	F, V	NI	Arm, hand, finger
Nagaraj e al., 2009 [58], Chiang et al., 2017 [59]	Nonwearable	3D controller	F, V, A	NI	Arm, hand
Sadihov et al., 2013 [60]	Wearable and nonwearable	Motion sensing camera (depth sensing, body tracking), data glove (bend sensing)	T, V	NI	Arm, hand, finger
Kapur et al., 2009 [61]	Wearable and nonwearable	Sleeve for optical tracking (camera)	T, V	NI	Arm
Ramírez-Fernández et al., 2015 [62]	Wearable and nonwearable	3D controller, motion sensing camera (depth sensing, hand tracking)	F, V, A	NI	Arm, hand

In studies using only visual feedback, auditory feedback could possibly be used. Abbreviations: I, immersive; NI, non-immersive; SI, semi-immersive; T, tactile; F, force; V, visual; A, auditory; EMG, electromyography; DOF, degrees of freedom; VR: virtual reality.

**Table 2 jcm-09-03369-t002:** Commercialized VR systems custom-built for upper limb rehabilitation.

VR System	VR Type	Sensor Type	Body Part	Company	Country
Riablo Premium	NI	IMU sensor	Arm	CoRehab	Italy
SaeboVR	NI	Motion-sensing camera (depth sensing, body tracking)	Arm	Saebo	USA
Doctor Kinetic	NI	Motion-sensing camera (depth sensing, body tracking)	Arm	Doctor Kinetic	Netherlands
IREX	NI	Motion sensing with webcam	Arm	GestureTek Health	Canada
Virtual Rehab	NI	Motion-sensing camera (depth sensing, body tracking, and hand tracking)	Arm, hand	Evolv	Spain
XR Health	I	HMD, controller	Arm	XR Health	USA
iWall	NI	Motion-sensing camera (depth sensing, body tracking), touch screen	Arm, hand	CSE Entertainment	Finland
Nirvana	NI	wall or floor touch sensing	Arm, hand	BTS Bioengineering	USA
Myro	NI	Touch screen, touchable objects on screen	Arm, hand	Tyromotion	USA
DIEGO	NI	Hand suspended type	Arm	Tyromotion	USA
AMADEO	NI	Position and force sensor in hand rehab robot	Finger	Tyromotion	USA
Pablo	NI	IMU sensor	Arm, hand	Tyromotion	USA
EsoGLOVE	NI	Hand exoskeleton	Arm, hand, finger	Roceso Technologies	Singapore
Bimeo PRO	NI	IMU sensor for body, IMU sensor in objects	Arm, hand	Kinestica	Slovenia
HandTutor	NI	Data glove	Hand, finger	Meditouch	Israel
Playball	NI	IMU sensor in ball	Hand	Tonkey	Italy
Anika	NI	Data glove	Hand, finger	ZARYA	Russia
Gloreha Workstation plus	NI	Hand exoskeleton, Optical sensor	Hand, finger	Gloreha	Italy
Icone	NI	Machine holding and moving handle	Arm	Heaxel	Italy
ExoRehab X	NI	Arm exoskeleton	Arm	HoustonBionic	Turkey
Hand of Hope	NI	Hand exoskeleton	Hand, finger	Rehab-Robotics Company	Hong Kong
SaeboRejoyce	NI	3D movable handle	Arm, hand	Saebo	USA
MindMotion Pro	NI	Colored object 3D tracking	Arm, hand	MindMaze	Switzerland
YouGrabber	NI	Data glove, infrared tracking camera	Arm, hand, finger	YouRehab	Switzerland
Rapel Smart Glove	NI	Data glove, IMU sensor	Arm, hand, finger	Neofect	South Korea
Smartboard	NI	2D handling board	Arm	Neofect	South Korea
MusicGlove	NI	Finger-to-finger contact	Finger	FlintRehab	USA
FitMi	NI	Puck with multiple sensors for movement tracking	Arm, hand	FlintRehab	USA
SensoRehab	NI	Data glove	Hand, finger	SensoMed	Russia
Rewellio	I	HMD, controller	Arm	Rewellio Inc.	USA

Abbreviations: I, immersive; NI, non-immersive; IMU, inertial measurement unit; HMD, head-mounted display; VR, virtual reality.

**Table 3 jcm-09-03369-t003:** Characteristics of the included meta-analyses.

Study	Aim	Search Strategy	Search Period	Inclusion Criteria	Included Trials, *n*	Participants, *n*
Lohse et al., 2014 [64]	To demonstrate the effect of virtual reality (VR) therapy among patients after stroke in both custom built virtual environments and commercial gaming systems.	MEDLINE, CINAHL, EMBASE, ERIC, PSYCInfo, DARE, PEDro, Cochrane Central Register of Controlled Trials, and Cochrane Database of Systematic Reviews	From inception to 4 April 2013	Randomized or quasi-randomized controlled trials with adults (>18 years old) after stroke, excluding other neurological disorders.	24	626
Laver et al., 2017 [68]	To investigate the efficacy of VR in comparison with alternative interventions or no interventions on the function and activity of hemiparetic upper limbs.	Cochrane Stroke Group Trials Register, CENTRAL, MEDLINE, Embase, and seven additional databases	From inception to April 2017	Randomized and quasi-randomized trials of VR rehabilitation in adults after stroke.	72	2470
Aminov et al., 2018 [63]	To review the evidence for VR in upper limb function and cognition after stroke.	Scopus, Cochrane Database, CINAHL, The Allied and Complementary Medicine Database, Web of Science, MEDLINE, Pre-Medline, PsycEXTRA, and PsycINFO	From inception to 28 June 2017	Randomized controlled trials utilizing a VR to improve either motor (upper limb) function, cognitive, or activities of daily living in patients with stroke.	31	971
Lee et al., 2019 [67]	To evaluate the effect of VR training on lower limb, upper limb, and overall functions in patients with chronic stroke.	OVID, PubMed, and EMBASE	From January 2000 to June 2018	Randomized controlled trials for using VR as a rehabilitation intervention in patients with chronic stroke.	21	562
Karamians et al., 2020 [66]	To demonstrate the efficacy of VR- and gaming-based rehabilitations for upper limb function in patients with stroke.	PubMed, CINAHL/EBSCO, SCOPUS, Ovid MEDLINE, and EMBASE	From 2005 to 2019	Randomized controlled trials or prospective study design with outcome measures of Wolf Motor Function Test, Fugl-Meyer Assessment or Action Research Arm Test in patients who had poststroke upper extremity deficits.	38	1198
Mekbib et al., 2020 [65]	To evaluate the therapeutic effect of VR compared to dose-matched conventional therapy in patients with stroke.	EMBASE, MEDLINE, PubMed, and Web of Science	From 2010 to February 2019	Randomized controlled trials that allocated patients either to a VR therapy or to a dose-matched conventional therapy.	27	1094

**Table 4 jcm-09-03369-t004:** Review of findings of included meta-analyses.

Study	Intervention	Comparison	Outcomes	Major Findings	Conclusions	Methodological Quality
Lohse et al., 2014 [64]	VR therapy: Custom-built VE or CG	CT	Behavioral assessment in body function, activity, or participation according to International Classification of Functioning (ICF)	(1) Body function -VE: SMD = 0.43, 95% CI = 0.22 to 0.64 -CG: SMD = 0.76, 95% CI = −0.17 to 1.70 (2) Activity -VE: SMD = 0.54, 95% CI = 0.28 to 0.81 -CG: SMD = 0.76, 95% CI = −0.25 to 1.76 (3) Participation -VE: SMD = 0.56, 95% CI = 0.02 to 1.10	VR rehabilitation moderately improves functional outcomes compared to CT in patients with stroke. CG studies were too few and small to evaluate the benefits of CG.	High
Laver et al., 2017 [68]	VR rehabilitation	Alternative intervention (usually CT) or no intervention	Upper limb function and activity	(1) Upper limb function (VR versus CT) -Composite: SMD = 0.07, 95% CI = −0.05 to 0.20 -FMA: SMD = 2.85, 95% CI = 1.06 to 4.65 (2) Upper limb function (additional VR) -Composite: SMD = 0.49, 95 CI = 0.21 to 0.77 (3) Activity of daily living -VR versus CT : SMD = 0.25, 95% CI = 0.06 to 0.43 -Additional VR : SMD = 0.44, 95% CI = 0.11 to 0.76	VR rehabilitation was not superior to CT in improving upper limb function. VR may be beneficial, when applied as an additional therapy to usual care, to improve the function of hemiparetic upper limbs and activities of daily living as additional VR therapy can increase overall therapy time.	High
Aminov et al., 2018 [63]	VR rehabilitation	CT	Upper limb function (e.g., FMA) and activity (e.g., BBT, BI) according to ICF	(1) Upper limb function : SMD = 0.41, 95% CI = 0.28 to 0.55 (2) Upper limb activity : SMD = 0.47, 95% CI = 0.34 to 0.60	VR can be beneficial on outcomes of body structure/function and activity in patients with stroke.	Moderate
Lee et al., 2019 [67]	VR rehabilitation	CT or no intervention	Upper limb function	(1) Upper limb function : SMD = 0.43, 95% CI = 0.42 to 0.54 (2) Lower limb function : SMD = 0.42, 95% CI = 0.34 to 0.51 (3) Overall function : SMD = 0.55, 95% CI = 0.25 to 0.84	VR training moderately improved function in patients with chronic stroke.	Low
Karamians et al., 2020 [66]	VR rehabilitation	CT or no intervention	Upper limb function (FMA, WMFT, ARAT)	(1) VR or gaming versus all controls : Percent possible improvement = 28.45%, 95% CI = 24.40 to 32.49% (2) VR or gaming versus CT : Percent possible improvement = 10.40%, 95% CI = 5.65 to 15.14%	VR- or gaming-based rehabilitation for upper limb function was more effective than CT in patients with stroke.	Moderate
Mekbib et al., 2020 [65]	VR rehabilitation	Dose-matched CT	Upper limb function (FMA, BBT, MAL)	(1) FMA : Mean difference = 3.84, 95% CI = 0.93 to 6.75 (2) BBT : Mean difference = 3.82, 95% CI = 0.26 to 7.38 (3) MAL : Mean difference = 0.80, 9% CI = 0.44 to 1.15	VR rehabilitation was more beneficial on post-stroke upper limb function in the outcomes of FMA, BBT and MAL than dose-matched CT.	Moderate

VR rehabilitation includes both rehabilitations using custom-built virtual environments and commercial video gaming consoles (e.g., Nintendo Wii or Xbox Kinect). FMA, BBT, WMFT, ARAT, and MAL are the measurement tools for upper limb function. Abbreviations: VR, virtual reality; VE, virtual environments; CG, commercially available gaming systems; CT, conventional therapy; ICF, International Classification of Functioning; SMD, standardized mean difference; CI, confidence interval; FMA, Fugl-Meyer Assessment; BBT, Box and Block Test; BI, Barthel Index; WMFT, Wolf Motor Function Test; ARAT, Action Research Arm Test; MAL, Motor Activity Log.

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
