# Peer review of "Clinical Application of Virtual Reality for Upper Limb Motor Rehabilitation in Stroke: Review of Technologies and Clinical Evidence"

_jcm, 2020, doi:10.3390/jcm9103369_

Round 1

Reviewer 1 Report

The manuscript constitutes a timely, interesting, and relevant topic to the continuously growing field of virtual reality in healthcare. While the manuscript offers detailed insight to several important aspects of virtual neurorehabilitation for stroke recovery, I believe that first and foremost, it would necessitate a more detailed section on the clinical relevance and findings/results of VR in motor recovery following stroke, and that it would also benefit from addressing the following points:

General comments

  • There are a few minor grammatical mistakes in the manuscript, please double check. It would be beneficial if a native English speaker read the manuscript as well.

Abstract

  • The primary focus of the manuscript is specific to the use of virtual reality (VR) for motor rehabilitation following stroke, so the opening sentence “neurorehabilitation for various diseases affecting the CNS” is a bit broad considering the specificity of the manuscript. Furthermore, stroke is not mentioned until quite late into the Abstract; I would suggest restructuring this section a bit. Considering that the readers of this journal have a primary interest in the clinical applications, the authors may consider emphasizing the importance of VR for stroke as a primary point earlier in the Abstract, followed by the technical details and advantages of VR therapy afterwards.
  1. Introduction
    • Stroke seems to be a leading cause of disability and socioeconomic burden, but not the (singular) leading cause of disability and socioeconomic burden.
    • Line 50: “However, the effect of conventional rehabilitation modalities is limited, and novel therapeutic approaches are required”: A brief introduction of some of the conventional rehabilitation approaches and their limitations would make an overall stronger point for the manuscript – e.g., mirror therapy: what are the limitations?
    • The term “gameification” is used in the Introduction section, but not explicated until much later in the manuscript (section 4.5). Readers of this journal may not be familiar with the term; therefore, it should be included earlier when first introduced.
    • Figure 1 would benefit from further elaboration/explication of each important component of neural plasticity, such as what “intensive” or a “step by step approach” refers to, and how VR addresses each of these points.
  2. Technologies used in VR rehabilitation

2.1 Definition of VR

  • The wording on line 75 “…and even recognize that they exist in the virtual environment” sounds a bit odd; why “even”? The presence in VR is easily realizable; “even” makes it sound like this constitutes a commendable feat.
  • It is correct that control of the avatar’s movement induces a sense of ownership of the virtual avatar, but it also and more specifically induces a sense of agency, which – together with body ownership – is an essential aspect of visuomotor-based virtual embodiment.
  • Similarly, it is questionable whether the virtual avatar truly constitutes an “extension” of their own body (line 82), as an extension would insinuate spatial lengthening; however, the avatar should ideally offer the same limb length as the physical body, but could in theory also be shorter. “…body parts as ‘surrogates’ of their own” may be a more applicable term.
  • Line 83 “…which allows them to experience where they are, with whom, and what they do there similar to the real/lived experience”: with whom comes as a bit of a surprise – do motor rehabilitation patients usually encounter another virtual avatar? The authors may consider removing sections such as this, if it does not specifically relate to neurorehabilitation.

2.2 Non-immersive and immersive VR

  • I would suggest that the authors consider restructuring this section to begin with non-immersive VR, since 1. The title begins with non-immersive VR, and 2. Building up from non-immersive to immersive VR could improve the flow of the manuscript, since immersive VR generally adds features (i.e., enhanced presence) compared to non-immersive VR.
  • Line 86: Certain word choices may not be quite accurate. For example, “immersive VR enables people to experience artificial environments as real life” – people arguably do not quite experience immersive VR as “real life” (they are, after all, fully aware that they are located in a virtual space), but immersive VR does, however, appropriate the presence that individuals experience in real life.
  • Line 87: “…while being completely isolated from the real surroundings” would not necessarily always be the case, as e.g., olfactory and auditory sensations from the outside world are still perceived while in VR, and most VR modalities employed in neurorehabilitation do not fully isolate the patient from external sensory experiences.
  • Line 95: The authors suggest that it “…might be helpful not to rely on a sophisticated artificial stimulator to provide sensations” for complete immersive VR: why would it be beneficial not to rely on such artificial stimulators? Wouldn’t stimulators enhance the sense of presence in VR and hence increase its efficacy?

2.3 Sensors and feedbacks for virtual rehabilitation

  • I would suggest inserting Table 1 towards the end of this section, as reading through the different sensor types is more confusing than if one would be able to read the content first (also, see comments below on creating a new table that summarizes the sensors for a better overview).
  • Considering the clinically-focused readership of this journal, the section on sensors and feedbacks for virtual rehabilitation reads quite technical, but also does not include all the necessary explanations of certain terminology.
  • The section jumps between motor, visual, auditory, and haptic feedback and could read as confusing to readers not well versed in the world of virtual reality. Judging by the data included in Table 1, most studies employed visuomotor (V) or visuotactile (F, H) congruency, as well as some auditory cues. However, since visuomotor (VM) and visuotactile (VT) congruency constitute the most widely employed means to enhance presence, I would suggest ordering this section according to these two differentiations. While some studies did include auditory cues, it seems that auditory feedback acted as complementary cues secondary to either VM or VT feedback.
  • Building on the point above, segmenting this section into 1. VM (motor) and 2. VT (haptic and force) feedback modalities would provide a better organization of the type of sensors that can be employed for each type.
  • However, several terms should be explicated in greater detail, since the readership of this journal is likely more clinically oriented rather than technologically versed. For example, explication of how a wearable and non-wearable device differ for recognition of motor movement (line 137), and along these lines, whether the “glove method or camera method” (line 147) would go in hand with wearable and non-wearable devices, respectively. (Also, see comments below for section 4.2 on segmenting the sensor types along “wearable” and “non-wearable” tracking devices)
  • The short section on VR for gait or balance rehabilitation mentions that the studies employed “body tracking using either a motion camera or motion sensor” (line 154), but no mention of haptic feedback is included here; however, it seems that at least some of the studies (e.g., [67]) used haptic feedback (treadmill with vibration feedback). When listing “treadmill”, does this refer to the users watching themselves walk in VR? Some of the differentiations are unclear, as it is stated that studies used a treadmill/cycling device and monitor gait through body tracking (motion camera/sensor): so, how does a “motion sensing camera” [61] differ from “Treadmill” (e.g., [69-70])? According to the definition above, treadmills are combined with motion sensing cameras, so the two would go hand in hand? This should be clarified.
  • Is there a specific difference as to why Table 1 uses “Sensor type” and Table 2 “Device type”? Along these lines, the tables should follow the same order in their headers. While Table 1 + 2 might differ in content, the overlapping parts should be in the same order, such as putting ‘VR type’ second to last consistently, rather than second to last in Table 1 and in second place in Table 2.  
  • Overall, tables 1 and 2 are packed with technical information. Would it be possible to include a broader sensor category, such as “camera/non-wearable” vs. “wearable” (see also comments to section 4.2)? The authors could certainly include an additional column that specifies the exact type of camera or wearable device that was used, since it is relevant information, but I believe that for the general readership of this journal, it would serve as a more easily understandable overview if “3D controller using buttons, 3D controller, glove with exoskeleton…” etc. could be grouped under “wearable tracking devices” and “motion sensing camera, marker-based tracking with webcam, colored object tracking with webcam…” etc. be grouped under “non-wearable/camera tracking devices”.

2.3 Sensors and feedbacks for virtual rehabilitation

  • This section could be elaborated on more. How do commercialized virtual rehabilitation devices differ from VR rehabilitation research? In what ways are they simplified and more focused on ease of use? Including specific examples would help shape the picture.
  • Furthermore, are these commercialized virtual rehabilitation devices for home use? Or only available in e.g., rehabilitation centers? Are there any downsides or reduced efficacy in using one over the other (commercialized versus VR rehab research)?
  1. Clinical evidence and consideration for VR in motor rehabilitation after stroke

3.2 Clinical evidence

  • Considering the suggested importance of differentiating between VR rehabilitation versus gaming-based rehabilitations, it would be vital to more clearly explain how these two approaches differ earlier in the manuscript, which would include a necessary clear definition of ‘gameification’ early on in the manuscript as well.
  • Importantly, considering the readership and focus of the journal, this section on the clinical evidence should constitute the most detailed of all sections in the entire manuscript, as it seems like it would be the heart of the manuscript and of greatest interest to its readers. However, I feel like it is not quite informative enough yet in its current form, and would necessitate a more thorough and detailed review and comparison of the existing research on the topic. For example, how did studies differ from one another – i.e., sensory feedback provided, immersive/non-immersive VR? This section on clinical evidence should be tied into the previous sections on immersive/non-immersive VR, and sensors and feedbacks, so that these technological explications of VR are applied to the clinical data.
  • Although Table 4 provides a very helpful overview, it would help the reader if the authors summarized the findings of Table 4 in more detail and connected the findings of these meta-analyses to the points addressed above (immersive nature of VR/sensor and feedbacks). What types of conclusions can be drawn based on the clinical data? E.g., is one type of sensory superior?
  • Furthermore, explication of abbreviations would be helpful to the general readership. Instead of stating that “FMA significantly improved”, it would be easier to follow for the general readership if the authors would instead describe what(as is measured by e.g., the FMA) improved significantly.
  1. Considerations for VR application in stroke rehabilitation

4.2 Movement visualization

  • Parts of this section again include terminology that readers of this journal may not be familiar with, such as “augmented reality”, or how “avatar MV” would differ from “tracking MV”, for example.
  • Furthermore, I feel like this section would fit better if integrated into section 2.3: sensors and feedbacks, especially if the authors organize section 2.3 along motor versus haptic feedback. If this were to be the case, section 4.2 on movement visualization would fit into the part on motor feedback and would therefore also constitute an informative introduction to the studies listed in the tables.
  • Overall, there is a lot of technical information packed into sections 2.3 and 4.2, especially when considering that many readers of the journal may not be familiar with VR. As an extension to my comments above on section 2.3, I would suggest that the authors create a table sectioned into 1. Motor feedback and 2. Haptic feedback, subdivided by “wearable” and “non-wearable” tracking devices, which can include all the different types of sensors employed in each. I believe it would be easier for readers of all backgrounds to follow if structured like this.
  • Line 246: why might there be such differences in skeleton tracking vs. optical tracking preference in patients vs. healthy volunteers/therapists?
  1. Summary
  • An interesting section may also be the potential use of VR in the hands of therapists versus as a tool for home use in patients’ hands directly.
  • It would be helpful if the authors added a few more detailed concluding remarks pertaining to the specificity of VR’s effectiveness in neurorehabiliation – i.e., is it more efficient when used with specific tracking (e.g., motor rather than haptic?) or more effective for upper versus lower limb?

Author Response

Thank you for your extensive review and valuable comments. We have endeavored to address all issues appropriately. Some of your comments, though also very valuable, were out of the range of our preplanned review. Our focus was to primarily give a general overview of VR applications, and some of your valuable comments were more specific and need more results from further research. Those things can be dealt with in a review with a greater scope, combining exploratory studies with small sample sizes, depending on the subjective judgement of experts. Therefore, please read our revision considering these things.

Comment 1: There are a few minor grammatical mistakes in the manuscript, please double check. It would be beneficial if a native English speaker read the manuscript as well.

Response: Thank you for your comment. We received professional editing from a native English speaker for our first manuscript, and we again used an editing service for this revision.

Comment 2: Abstract The primary focus of the manuscript is specific to the use of virtual reality (VR) for motor rehabilitation following stroke, so the opening sentence “neurorehabilitation for various diseases affecting the CNS” is a bit broad considering the specificity of the manuscript. Furthermore, stroke is not mentioned until quite late into the Abstract; I would suggest restructuring this section a bit. Considering that the readers of this journal have a primary interest in the clinical applications, the authors may consider emphasizing the importance of VR for stroke as a primary point earlier in the Abstract, followed by the technical details and advantages of VR therapy afterwards.

Response: Thank you for this valuable comment. We made efforts to revise the Abstract to meet both your and another reviewer’s comments, while keeping the Abstract within the 200-word limit (lines 21-35).

Introduction

Comment 3: Stroke seems to be leading cause of disability and socioeconomic burden, but not the(singular) leading cause of disability and socioeconomic burden.

Response: Thank you for your comment. We have revised this sentence to “Stroke is one of the leading causes of disability and socioeconomic burden worldwide.” (Please see line 40.)

Comment 4: Line 50: “However, the effect of conventional rehabilitation modalities is limited, and novel therapeutic approaches are required”: A brief introduction of some of the conventional rehabilitation approaches and their limitations would make an overall stronger point for the manuscript – e.g., mirror therapy: what are the limitations?

Response: This sentence is the consensus comment in this field. The authors did not feel much need to describe the problems of each therapy for upper limb rehabilitation in this paragraph. In the following paragraph (lines 52-65), we have suggested the possible benefit of VR rehabilitation when added to conventional rehabilitation.

Comment 5: The term “gameification” is used in the Introduction section, but not explicated until much later in the manuscript (section 4.5). Readers of this journal may not be familiar with the term; therefore, it should be included earlier when first introduced.

Response: The authors thought that the word “gamification” may be familiar to general readers, as gamification is used in the general field (e.g., in education, marketing, medicine, and so on). However, we have added the following definition of gamification in the Introduction (lines 58-59): “the process of adding game-design elements and game principles to something (e.g. task) so as to encourage participation.”

Comment 6: Figure 1 would benefit from further elaboration/explication of each important component of neural plasticity, such as what “intensive” or a “step by step approach” refers to, and how VR addresses each of these points.

Response: Thank you for your comment. We have added more explanation to the corresponding paragraph (lines 52-65).

Technologies used in VR rehabilitation

2.1 Definition of VR

Comment 6:  The wording on line 75 “…and even recognize that they exist in the virtual environment” sounds a bit odd; why “even”? The presence in VR is easily realizable; “even” makes it sound like this constitutes a commendable feat.

Response: Thank you for your comment. This was a mistake in writing. We have revised the sentence to convey the meaning more clearly. (See lines 75-79.)

Comment 7:  It is correct that control of the avatar’s movement induces a sense of ownership of the virtual avatar, but it also and more specifically induces a sense of agency, which – together with body ownership – is an essential aspect of visuomotor-based virtual embodiment. Similarly, it is questionable whether the virtual avatar truly constitutes an “extension” of their own body (line 82), as an extension would insinuate spatial lengthening; however, the avatar should ideally offer the same limb length as the physical body, but could in theory also be shorter. “…body parts as ‘surrogates’ of their own” may be a more applicable term.

Response: Thank you for your kind suggestion. We have replaced the word “extensions” with “surrogates” (line 85).

Comment 8:  Line 83 “…which allows them to experience where they are, with whom, and what they do there similar to the real/lived experience”: with whom comes as a bit of a surprise – do motor rehabilitation patients usually encounter another virtual avatar? The authors may consider removing sections such as this, if it does not specifically relate to neurorehabilitation.

Response: Thank you for your comment. As you suggested, the sentence was originally about VR in general, not for specific fields, such as neurorehabilitation. We have revised the sentence by omitting “with whom” (lines 86-87) so that the meaning does not imply what does not specifically relate to neurorehabilitation.

2.2 Non-immersive and immersive VR

Comment 9:  I would suggest that the authors consider restructuring this section to begin with non-immersive VR, since 1. The title begins with non-immersive VR, and 2. Building up from non-immersive to immersive VR could improve the flow of the manuscript, since immersive VR generally adds features (i.e., enhanced presence) compared to non-immersive VR.

Response: Thank you for your suggestion. We have reordered these paragraphs so that non-immersive VR is addressed first.

Comment 10:  Line 86: Certain word choices may not be quite accurate. For example, “immersive VR enables people to experience artificial environments as real life” – people arguably do not quite experience immersive VR as “real life” (they are, after all, fully aware that they are located in a virtual space), but immersive VR does, however, appropriate the presence that individuals experience in real life.

Response: We agree with your opinion. We have changed this sentence following your advice (lines 103-106).

.

Comment 11:  Line 87: “…while being completely isolated from the real surroundings” would not necessarily always be the case, as e.g., olfactory and auditory sensations from the outside world are still perceived while in VR, and most VR modalities employed in neurorehabilitation do not fully isolate the patient from external sensory experiences.

Response: We agree with your opinion. We have changed this sentence following your advice lines (103-106).

Comment 12:  Line 95: The authors suggest that it “…might be helpful not to rely on a sophisticated artificial stimulator to provide sensations” for complete immersive VR: why would it be beneficial not to rely on such artificial stimulators? Wouldn’t stimulators enhance the sense of presence in VR and hence increase its efficacy?

Response: Yes, you are correct. This may be a language problem. We have revised this sentence to reveal our original intent, as follows: “These could be further covered by complete immersive VR, but the need for a sophisticated artificial stimulator to provide variable sensations may require more space and have a higher cost.” (Please see lines 111-113.)

2.3 Sensors and feedbacks for virtual rehabilitation

Comment 12:  I would suggest inserting Table 1 towards the end of this section, as reading through the different sensor types is more confusing than if one would be able to read the content first (also, see comments below on creating a new table that summarizes the sensors for a better overview).

Response: Thank you for noting this. We have moved Table 1 to the end of this section.

Comment 13:  Considering the clinically-focused readership of this journal, the section on sensors and feedbacks for virtual rehabilitation reads quite technical, but also does not include all the necessary explanations of certain terminology.The section jumps between motor, visual, auditory, and haptic feedback and could read as confusing to readers not well versed in the world of virtual reality. Judging by the data included in Table 1, most studies employed visuomotor (V) or visuotactile (F, H) congruency, as well as some auditory cues. However, since visuomotor (VM) and visuotactile (VT) congruency constitute the most widely employed means to enhance presence, I would suggest ordering this section according to these two differentiations. While some studies did include auditory cues, it seems that auditory feedback acted as complementary cues secondary to either VM or VT feedback. Building on the point above, segmenting this section into 1. VM (motor) and 2. VT (haptic and force) feedback modalities would provide a better organization of the type of sensors that can be employed for each type.

Response: Thank you for your comments. We divided the studies by visuomotor feedback use and visuohaptic feedback use within Table 1. Haptic feedback includes tactile feedback and force feedback depending on whether resistance is present. Thus, we used the word “visuohaptic” rather than “visuotactile.” We have described this in more detail in the corresponding paragraph (lines 153-166).

Comment 14:  However, several terms should be explicated in greater detail, since the readership of this journal is likely more clinically oriented rather than technologically versed. For example, explication of how a wearable and non-wearable device differ for recognition of motor movement (line 137), and along these lines, whether the “glove method or camera method” (line 147) would go in hand with wearable and non-wearable devices, respectively. (Also, see comments below for section 4.2 on segmenting the sensor types along “wearable” and “non-wearable” tracking devices)

Response: We agree with your comments. We have explained these terms in greater detail to ensure the understanding of clinically oriented readers. (Please see Section 2.3.)

Comment 15:  The short section on VR for gait or balance rehabilitation mentions that the studies employed “body tracking using either a motion camera or motion sensor” (line 154), but no mention of haptic feedback is included here; however, it seems that at least some of the studies (e.g., [67]) used haptic feedback (treadmill with vibration feedback). When listing “treadmill”, does this refer to the users watching themselves walk in VR? Some of the differentiations are unclear, as it is stated that studies used a treadmill/cycling device and monitor gait through body tracking (motion camera/sensor): so, how does a “motion sensing camera” [61] differ from “Treadmill” (e.g., [69-70])? According to the definition above, treadmills are combined with motion sensing cameras, so the two would go hand in hand? This should be clarified.

Response: Thank you for your comments. Considering that the focus on this review was upper limb rehabilitation, in line with the comments of Reviewer #2, we have clearly stated that in the title and Abstract, and then removed this section and Table 2.

Comment 16:  Is there a specific difference as to why Table 1 uses “Sensor type” and Table 2 “Device type”? Along these lines, the tables should follow the same order in their headers. While Table 1 + 2 might differ in content, the overlapping parts should be in the same order, such as putting ‘VR type’ second to last consistently, rather than second to last in Table 1 and in second place in Table 2.  

Response: As noted in the response to Comment 15, we have removed this paragraph and Table 2.

Comment 17:  Overall, tables 1 and 2 are packed with technical information. Would it be possible to include a broader sensor category, such as “camera/non-wearable” vs. “wearable” (see also comments to section 4.2)? The authors could certainly include an additional column that specifies the exact type of camera or wearable device that was used, since it is relevant information, but I believe that for the general readership of this journal, it would serve as a more easily understandable overview if “3D controller using buttons, 3D controller, glove with exoskeleton…” etc. could be grouped under “wearable tracking devices” and “motion sensing camera, marker-based tracking with webcam, colored object tracking with webcam…” etc. be grouped under “non-wearable/camera tracking devices”.

Response: As noted in the response to Comment 15, we have removed Table 2. To improve understanding, as you commented, a new column was added to Table 1, and we have divided the sensor types into wearable and non-wearable devices.

2.3 Sensors and feedbacks for virtual rehabilitation

Comment 18:  This section could be elaborated on more. How do commercialized virtual rehabilitation devices differ from VR rehabilitation research? In what ways are they simplified and more focused on ease of use? Including specific examples would help shape the picture. Furthermore, are these commercialized virtual rehabilitation devices for home use? Or only available in e.g., rehabilitation centers? Are there any downsides or reduced efficacy in using one over the other (commercialized versus VR rehab research)?

Response: Thank you for your valuable comment. The purpose of the summary of commercialized devices was to introduce the readers to the possibilities currently available for VR rehabilitation systems in the clinical setting. As you know, some of the clinical trials used their prototype in the rehabilitation, and others used commercialized devices. We think that comparing the prototype used in the research and commercialized devices is outside the focus in this review. The commercialized VR systems in Table 2 are different from the commercialized gaming systems not specifically designed for rehabilitation (e.g., Nintendo Wii, Xbox Kinect). We have changed the title of Table 2 to “Commercialized VR systems custom-built for upper limb rehabilitation.” Additionally, we addressed this issue in the clinical evidence section according to Comment #19 (custom-built VR systems [irrespective of being commercialized or not] vs. non-custom-built gaming systems [usually commercialized devices, such as Nintendo Wii an Xbox Kinect]). Please see Section 2.3.

The home- or center-use for commercialized devices was also outside the focus of this review. Usually, light and portable systems are developed to be used in both home and hospital settings, but the decision is complex, according to the healthcare payment systems, providers’ or consumers’ preferences, and so on. More bulky and expensive systems are usually difficult to use in the home setting due space and money limitations.

Clinical evidence and consideration for VR in motor rehabilitation after stroke

3.2 Clinical evidence

Comment 19:  Considering the suggested importance of differentiating between VR rehabilitation versus gaming-based rehabilitations, it would be vital to more clearly explain how these two approaches differ earlier in the manuscript, which would include a necessary clear definition of ‘gameification’ early on in the manuscript as well.

Response: Thank you for your comment. All meta-analyses included both custom-built and commercialized gaming systems. We have more clearly stated this in the second paragraph of Section 3.2: “VR rehabilitation included rehabilitations using both custom-built virtual environments and commercial video gaming consoles (e.g., Nintendo Wii or Xbox Kinect) in the selected meta-analyses.” In addition, we have added the definition of “gamification” in the Introduction, as noted in our response to Comment 5 (lines 58-59).

Comment 20:  Importantly, considering the readership and focus of the journal, this section on the clinical evidence should constitute the most detailed of all sections in the entire manuscript, as it seems like it would be the heart of the manuscript and of greatest interest to its readers. However, I feel like it is not quite informative enough yet in its current form, and would necessitate a more thorough and detailed review and comparison of the existing research on the topic. For example, how did studies differ from one another – i.e., sensory feedback provided, immersive/non-immersive VR? This section on clinical evidence should be tied into the previous sections on immersive/non-immersive VR, and sensors and feedbacks, so that these technological explications of VR are applied to the clinical data.

Response: Thank you for your comments. You pointed out the important issues in VR rehabilitation. However, we could not find sound evidence for those questions. For example, there are few head-to-head studies comparing immersive vs. non-immersive VR. The number of well-designed clinical trials using immersive VR is too small compared to those using non-immersive VR. As the number of studies in this field expands, it may be possible to determine more information about the types of VR that are likely to be more effective.

For the sensors, we cannot conclude which type of sensor is more effective for motor recovery. The authors do not think that the sensor itself can substantially affect the effect of VR rehabilitation. The sensor types can be determined based on the preferences of patients or therapists, durability, costs, purpose of rehabilitation, body segments to be tracked, and so on.

Comment 21:  Although Table 4 provides a very helpful overview, it would help the reader if the authors summarized the findings of Table 4 in more detail and connected the findings of these meta-analyses to the points addressed above (immersive nature of VR/sensor and feedbacks). What types of conclusions can be drawn based on the clinical data? E.g., is one type of sensory superior?

Response: Along with Comment 20, these comments are very valuable. However, we cannot draw conclusions for these questions based on the current published data. Please note that our review was not preplanned for these issues. The authors think that these issues could be discussed in another narrative scoping review by other experts.

Comment 22:  Furthermore, explication of abbreviations would be helpful to the general readership. Instead of stating that “FMA significantly improved”, it would be easier to follow for the general readership if the authors would instead describe what(as is measured by e.g., the FMA) improved significantly.

Response: Thank you for your comment. The authors think that the descriptions in the second paragraph of Section 3.2 can help the readers follow the meaning of FMA and so on: “Outcomes were usually the composite outcomes of upper limb function or activities, and Karamians et al. only included studies using one of the following outcome measures: Fugl-Meyer Assessment (FMA), Wolf Motor Function Test (WMFT), and Action Research Arm Test (ARAT) [66]. Mekbib et al. only included studies using one of the three following outcomes: FMA, Box and Block Test (BBT), and Motor Activity Log (MAL) [65].” (See lines 202-206.) We also clearly stated that those measurement tools are designed for upper limb function in the manuscript and table 4.

Considerations for VR application in stroke rehabilitation

4.2 Movement visualization

Comment 22:  Parts of this section again include terminology that readers of this journal may not be familiar with, such as “augmented reality”, or how “avatar MV” would differ from “tracking MV”, for example.Furthermore, I feel like this section would fit better if integrated into section 2.3: sensors and feedbacks, especially if the authors organize section 2.3 along motor versus haptic feedback. If this were to be the case, section 4.2 on movement visualization would fit into the part on motor feedback and would therefore also constitute an informative introduction to the studies listed in the tables. Overall, there is a lot of technical information packed into sections 2.3 and 4.2, especially when considering that many readers of the journal may not be familiar with VR. As an extension to my comments above on section 2.3, I would suggest that the authors create a table sectioned into 1. Motor feedback and 2. Haptic feedback, subdivided by “wearable” and “non-wearable” tracking devices, which can include all the different types of sensors employed in each. I believe it would be easier for readers of all backgrounds to follow if structured like this.

Response: Thank you for your comments. We have removed this section, and some sentences in this part were moved to the portion discussing sensors (Section 2.3). For additional relevant responses, please see earlier in this letter.

Comment 23:  Line 246: why might there be such differences in skeleton tracking vs. optical tracking preference in patients vs. healthy volunteers/therapists?

Response: We removed this sentence during the revision, moving content from Section 4.2 to the sensor part (Section 2.3). In this study, they just showed that the accuracy was different among three types of sensors, but the preferences were different regardless of the accuracy of the tracking. In addition, sensor preferences were different between patients and therapists. However, they did not perform a direct survey for this reason in patients and therapists. In addition to accuracy, the ease of use, costs, and other factors will affect preferences.

Summary

Comment 24:  An interesting section may also be the potential use of VR in the hands of therapists versus as a tool for home use in patients’ hands directly. It would be helpful if the authors added a few more detailed concluding remarks pertaining to the specificity of VR’s effectiveness in neurorehabiliation – i.e., is it more efficient when used with specific tracking (e.g., motor rather than haptic?) or more effective for upper versus lower limb?

Response: Thank you for your comment. The authors think that the current research is not sufficient to draw conclusions to these questions. Although, VR can be a useful tool for home-based rehabilitation, and some clinical trials have revealed the possibility (as we shortly discussed in the combinational approach section [Section 5]), we think that it is not possible to discuss superiority. The authors think that the current level of evidence showed us that VR-based home-based rehabilitation can be a noninferior option to conventional center-based rehabilitation therapy. For sensors and tracking methods, the specific types used are decided according to the purpose of the systems, which may not lead to superiority in the efficacy or effectiveness of VR rehabilitation systems. In addition, we cannot say whether VR is more useful for upper limb rehabilitation than lower limb rehabilitation. We have briefly stated the limitation of current published research and suggested the direction for future research in the Summary (which is also in accordance with the comments of Reviewer #2).

Reviewer 2 Report

The work addresses an interesting and important topic relevant to clinical applications and trials of VR for motor rehabilitation after the stroke. I think that the manuscript is not ready at this time, but encourage the authors to consider the comments offered, and to make revisions to strengthen the manuscript.

In abstract, i think it would be appropriate to reduce the introduction and add the methods used for bibliographic research. Also, specify that the revision focuses on the upper limb.

Line 45: “disabilities” not “disability”.

Line 72-76. You should make the sentence clearer, I think it is too long.

In the VR section, it could be useful to consider these studies:

-Manuli A, Maggio MG, Latella D, Cannavò A, Balletta T, De Luca R, Naro A, Calabrò RS. Can robotic gait rehabilitation plus Virtual Reality affect cognitive and behavioural outcomes in patients with chronic stroke? A randomized controlled trial involving three different protocols. J Stroke Cerebrovasc Dis. 2020 Aug;29(8):104994. doi: 10.1016/j.jstrokecerebrovasdis.2020.104994.

-Calabrò RS, Accorinti M, Porcari B, Carioti L, Ciatto L, Billeri L, Andronaco VA, Galletti F, Filoni S, Naro A. Does hand robotic rehabilitation improve motor function by rebalancing interhemispheric connectivity after chronic stroke? Encouraging data from a randomised-clinical-trial. Clin Neurophysiol. 2019 May;130(5):767-780. doi: 10.1016/j.clinph.2019.02.013.

Line 168.   In the Literature search, please indicate who reviewed the documents (i.e. initials of authors).  Did the same authors review all papers? How were discrepant opinions handled?

Line 392. Also, enter the limits of the reviewed studies and the future prospects considered by the authors.

Author Response

Thank you for your extensive review and valuable comments. We endeavored to address all issues appropriately.

Comments 1: In abstract, i think it would be appropriate to reduce the introduction and add the methods used for bibliographic research. Also, specify that the revision focuses on the upper limb.

Response: Thank you for this valuable comment. We have made efforts to revise the Abstract to meet both your and another reviewer’s comments within the word limit (200 words).

Comments 2: Line 45: “disabilities” not “disability”.

Response: Thank you. We have changed the word “disability” to “disabilities” (line 46).  

Comments 3: Line 72-76. You should make the sentence clearer, I think it is too long.

Response: Thank you. We have revised this long sentence as follows: “VR technology can give users the experience of being surrounded by a computer-generated world. With VR, uses experience inclusive and, extensive surroundings, with vivid illusions of a virtual computer-generated environment in which, both realistic and unrealistic events can occur. So users can interact as though they were in a real environment, and may not even recognize that they are existing in a virtual environment [19].” (See lines 75-79.)

Comments 4: In the VR section, it could be useful to consider these studies:

-Manuli A, Maggio MG, Latella D, Cannavò A, Balletta T, De Luca R, Naro A, Calabrò RS. Can robotic gait rehabilitation plus Virtual Reality affect cognitive and behavioural outcomes in patients with chronic stroke? A randomized controlled trial involving three different protocols. J Stroke Cerebrovasc Dis. 2020 Aug;29(8):104994. doi: 10.1016/j.jstrokecerebrovasdis.2020.104994.

-Calabrò RS, Accorinti M, Porcari B, Carioti L, Ciatto L, Billeri L, Andronaco VA, Galletti F, Filoni S, Naro A. Does hand robotic rehabilitation improve motor function by rebalancing interhemispheric connectivity after chronic stroke? Encouraging data from a randomised-clinical-trial. Clin Neurophysiol. 2019 May;130(5):767-780. doi: 10.1016/j.clinph.2019.02.013.

Response: Thank you for your suggestion. We have added the study by Calabro et al. to Table 1. We removed Table 2 and the corresponding paragraph to focus more the upper limb rehabilitation.

Comments 5: Line 168.   In the Literature search, please indicate who reviewed the documents (i.e. initials of authors). Did the same authors review all papers? How were discrepant opinions handled?

Response: Although our review was not a formal umbrella review for systematic reviews, we partially followed the usual steps for this type of review. Our focus was to find the meta-analyses for VR upper limb rehabilitation after stroke, as the process was easier. The initial screening for all possible full papers written in English was conducted by Y.K. Then, two authors (Y.K. and W.S.K.) selected meta-analyses for review, among 339 studies. The quality appraisals for six selected meta-analyses were conducted by two authors (S.C. and W.S.K.). We have added this information to Sections 3.1 and 3.2.

Comments 6: Line 392. Also, enter the limits of the reviewed studies and the future prospects considered by the authors.

Response: Thank you for your comment. We have added the limits of the reviewed studies and suggestions for future clinical trials in the Summary: “VR-based rehabilitation is a promising tool to actively engage patients in the rehabilitation program, and it can lead to better motor recovery. Although current clinical evidence shows that VR-based rehabilitation is beneficial as an adjunct therapy to convenient rehabilitation therapy, the interventions in the studies included in the metanalyses were heterogeneous, and it is unclear who benefits more from VR rehabilitation (e.g., severity of impairment, time since onset of stroke) and what type of VR (e.g., immersive vs. non-immersive) and feedback is more effective. Further research, including large, well-designed RCTs, to find the factors influencing the effects of VR rehabilitation are required.” (See lines 373-380.)

Round 2

Reviewer 1 Report

Thank you, I have no additional comments.